# Ontologue: Declarative Benchmark Construction for Hierarchical Multi-Label Classification

**Sean T. Yang**
Electrical and Computer Engineering
University of Washington
Seattle, WA 98195
tyyang38@uw.edu

**Bernease Herman**
Information School
University of Washington
Seattle, WA 98195
bernease@uw.edu

**Bill Howe**
Information School
University of Washington
Seattle, WA 98195
billhowe@uw.edu

## Abstract

We describe a customizable benchmark for hierarchical and ontological multi-label classification, a task where labels are equipped with a graph structure and data items can be assigned multiple labels. We find that current benchmarks do not adequately represent the problem space, casting doubt on the generalizability of current results. We consider three dimensions of the problem space: context (availability of rich features on the data and labels), distribution of labels over data, and graph structure. For context, the lack of complex features on the labels (and in some cases, the data) artificially prevent the use of modern representation learning techniques as an appropriate baseline. For distribution, we find the long tail of labels over data constitute a few-shot learning problem that artificially confounds the results: for most common benchmarks, over 40% of the labels have fewer than 5 data points in the training set. For structure, we find that the correlation between performance and the height of the tree can explain some of the variation in performance, informing practical utility. In this paper, we demonstrate how the lack of diversity in benchmarks can confound performance analysis, then present a declarative query system called Ontologue for generating custom benchmarks with specific properties, then use this system to design 4 new benchmarks extracted from DBPedia that better represent the problem space. We evaluate state-of-the-art algorithms on both existing and new benchmarks and show that the performance conclusions can vary significantly depending on the dimensions we consider. We intend the system and derived benchmarks to improve the analysis of generalizability for these problems.

## 1 Introduction

In high-expertise domains, ontologies are used to capture expert knowledge and model a universe of discourse. For example, the Gene Ontology [1] [1] offers a logical structure describing complexity of the biological systems, from the molecular level to larger pathways, cellular and organism-level systems. It provides a computational representation of the current knowledge of genes from organisms. DBPedia [2] [2] hosts an open knowledge graph (OKG) modeling all concepts in various Wikimedia projects through a crowd-sourced community effort. WordNet [34] and ImageNet [13] are other examples of hierarchical- and graph-structured knowledge systems for natural language processing and computer vision research, respectively. Although broadly used, these ontological relationships are often ignored in machine learning settings. For example, AlexNet [24], ResNet [18], and VGG [40], popular benchmarks for computer vision applications, disregard the structure among labels.

---

[1] http://www.geneontology.org
[2] https://www.dbpedia.org/

An emerging community aims to use the label relationships as a source of constraints and sometimes supervision for multi-label learning. Hierarchical multi-label classification (HMC) considers classification with multiple labels per data item, selecting from a large and potentially evolving label space, while respecting hierarchical constraints: data items are labeled with a set of internal and leaf nodes, such that if a label is assigned, so are its ancestors. The HMC problem is especially relevant in high-throughput data generation settings, such as functional genomics [17, 38] and social media [49], where automating the labeling process requires high expertise but must be automated for scalability.

The development of HMC algorithms [45, 17, 49] has been enabled by the availability of about 20 common benchmark datasets. These datasets consist of protein function prediction [12, 42], annotation of medical images [14, 15], or text classification [23, 28]. While these datasets have helped inform new methods, we show that as a set they are generally small, self-similar, and can produce misleading performance results: they encode a bi-modality of an artificially challenging few-shot learning problem and an artificially easy multi-label classification problem, such that even a naive binary cross entropy solution can outperform all but the state of the art methods. In particular, we find that over 40% of labels have less than 5 data points in 16 of the 20 benchmark datasets, which constitutes a few-shot learning challenge beyond even those datasets designed to study few-shot learning [19]! Besides encouraging misleading conclusions, these datasets are artificially impoverished such that we cannot even apply state of the art few-shot learnign solutions as a baseline: the rich text associated with data items have been pre-processed into a fixed set of simple numeric features, preventing the use of modern representation learning techniques. The text associated with the label space has also been stripped away, preventing meaningful comparison with methods that can take advantage of it as a source of supervision. Moreover, not all HMC applications constitute few-shot learning problems, so we are arguably studying the wrong problem.

We present Ontologue , a toolkit for constructing benchmarks for graph- and hierarchy-based multi-label classification problems, along with four new benchmarks derived using the system, as an evaluation tool for HMC researchers. Ontologue provides a simple declarative query interface for DBPedia [2], allowing indirect influence over the application context (any topic in Wikipedia can be used as the root), the distribution of labels and data (each label included must be associated with a minimum number of data items), and the structure of the graph (the number of hops can be specified to control the height of the hierarchy.) The resulting benchmarks represent a hierarchical (or ontological) multi-label classification task involving the assignment of labels to relevant Wikipedia abstracts. The resulting benchmarks are large, interpretable, customizable, and most importantly, exhibit statistical and structural properties more suitable for performance analysis of state of the art methods: more balanced distribution of labels and data, fewer degenerate labels that are either trivial or impossible, availability of rich text features on both data and labels to afford multi-modal techniques, a greater proportion of DAG (Directed Acyclic Graph) structures (labels with multiple parents), and a more balanced distribution of labels over the height of the tree.

The following is a summary of our contributions in this paper:

- We analyze 20 HMC benchmark datasets and study several weaknesses in their utility as benchmark datasets due to structural and statistical properties.

- We propose a toolkit Ontologue to declaratively construct and analyze a subset from DBpedia as a benchmark for HMC problems with indirect control over structural and statistical features.

- We derive four new benchmarks using Ontologue and evaluate the performance of state of the art models using them, finding that they can change the order of state of the art results, while affording new multi-modal methods that are superior.

- We perform a comparative analysis of the new benchmarks against the old benchmarks and find that their more realistic statistical and structural properties can explain the differences in performance, suggesting new insights into the behavior of HMC methods.

## 2   Background and Related Work

We introduce the HMC problem and review existing methods and benchmarks used in its study.

**Hierarchical Multi-label Classification**   The HMC problem setting can be defined as follows: Given data items $X$ and a label graph $\mathcal{G} = (\mathcal{V}, \mathcal{E})$, find a function $f : X \rightarrow \mathcal{P}(\mathbb{V})$ mapping each

data item to a set of labels from the graph. The data items $X$ can be text, images, feature vectors, or embeddings. Although typically the graphs are considered trees, they may in some cases be DAGs, arbitrary graphs, or rich ontologies. In some settings, the goal is to constrain $f$ to obey the hierarchy such that if $c$ is in $f(x)$, the parents of $c$ are also in $f(x)$.

HMC methods can be categorized into global approaches and local approaches. Global approaches are designed to model the entire hierarchy [42, 5, 31], while local approaches split the problem into multiple classification tasks by level [7, 8, 9, 29, 51], or by node [11, 16, 4, 25, 47]. With the advance of deep learning in the last decade, neural networks have been used successfully to handle HMC problems. Wehrmann et al. [45] proposed HMCN-F and HMCN-R which can be seen as a hybrid approach, because they are simultaneously optimized by a global loss and a local loss. HMCN-F is a feed-forward layer and each layer represents one hierarchy level. While HMCN-F produces better results than HMCN-R (a recurrent network), the parameters of HMCN-F increase significantly as the hierarchy grows for higher computation time. C-HMCNN, presented by Giunchiglia et al. [17], produces predicted outputs that follow hierarchy constraint by introducing a modified binary cross entropy loss to enforce predicted probabilities of a decedent node to be lower than its ancestors. Surj [49] uses a graph neural network to learn a representation of each label and optimizes the network by mapping data items onto the representation space with cosine similarity. Nearly all of these studies evaluate their algorithms on some subset of 20 common benchmarks; we show the inadequacy of these benchmarks in Section 4.

**Entity Typing**    Entity typing [35, 37] has been a popular problem in the natural language processing community and it shares similar problem settings with Ontologue : the input is often sentences and the expected outputs are pre-determined categories. Despite similar setups, there are significant distinctions between entity typing problem and datasets generated with Ontologue. Medmentions [35] and TypeNet [37] are two examples of entity typing datasets. They are focusing on three tasks: Mention-Level Typing, Entity-Level Typing, and Entity Linking. The research MedMentions and TypeNet are facilitating is leveraging entity linking to categorize input sentences. HMC datasets can be seen as a Mention-Level Typing task with hierarchical constraints: Given a sentence vector, the model assigns multi-label while these labels follow the hierarchy structure. Another distinction is that the categories for MedMentions and TypeNet were derived from the entities included in the sentences, while the categories for Ontologue are assigned directly to the inputs.

**Existing Benchmarks**    While the HMC problem manifests in a wide variety of settings, the benchmark datasets are primarily drawn from a few specific application domains, including protein function prediction [12, 42], annotation of medical images [14, 15], and email text classification [23]. *Enron* was introduced in 2004 by Klimt et al. [23]. The inputs are feature vectors, including individual sections (From, To, Subject, and Body), and the outputs are user-specific folders. *FUNCAT* and *GO* collections are a collaborative work by Vens et al. [42] and Clare et al. [12]. The datasets within these two collections describe different aspects of the genes in yeast genome. The attributes include five types of bioinformatic data: sequence statistics, phenotype, secondary structure, homology, and expression. Both collections have identical input attributes, but the classes are taken from different knowledge graphs. FunCat is organized by the tree-structured MIPS [33] to classify the functions of gene products. The Gene Ontology (*GO*) [1] is provides a DAG-structured class hierarchy, allowing terms to have multiple parents. *ImCLEF07A*, *ImCLEF07D* [14], and *Diatoms* [15] are image classification datasets introduced by Dimitrovski et al. *ImCLEF07A* and *ImCLEF07D* include Local Binary Pattern (LBP) and Edge Histogram Descriptors (EHD) for texture features and Scale-Invariant Feature Transform (SIFT) for local features of X-Ray images, but do not provide the original images. The annotations of these two datasets are part of the IRMA hierarchical classification scheme [27]. *Diatoms* is a dataset to classify images of diatoms into hierarchical taxonomic rank. The feature attributes are product of Fourier descriptors and SIFT descriptors.

## 3   Ontologue Toolkit

Ontologue is a toolkit (Figure 1) for ontological multi-label classification dataset construction from DBPedia. This toolkit allows users to control contextual, distributional, and structured properties and create customized datasets. We also further provide four benchmark datasets that better represent the problem space from the toolkit and implement the existing HMC algorithms to provide baselines.

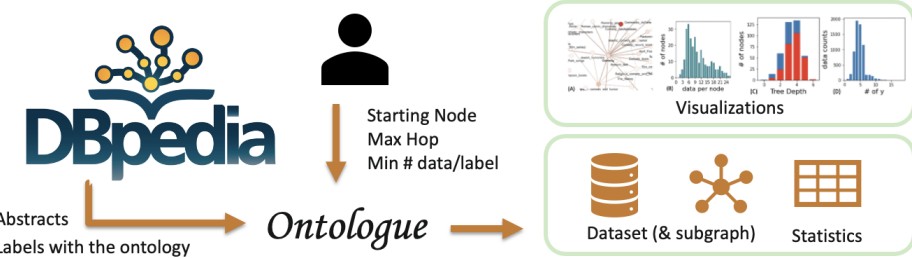

Figure 1: Pipeline of Ontologue . Ontologue is a toolkit for ontological multi-label classification dataset construction from DBPedia. This toolkit allows users to control contextual, distributional, and structured properties and create customized datasets. Ontologue also produces associated statistics and visualizations for the user to analyze and assess. We also further provide four benchmark datasets that better represent the problem space from the toolkit and implement the existing HMC algorithms to provide baselines.

## 3.1  Benchmark Requirements

We design Ontologue to offer declarative influence over three properties that can influence performance of various methods. We intentionally designed Ontologue to limit explicit, low-level control over benchmark properties to discourage the creation of cherry-picked datasets that would invalidate generalizability claims.

**Application Context**  Existing benchmark datasets are of a fixed and sometimes opaque domain, complicating interpretation and preventing the use of multi-modal methods. For example, the Gene Ontology datasets have been stripped of information: node labels have been replaced with integers, and data item features may be pre-processed using outdated feature engineering techniques. While this approach is reasonable to isolate and demonstrate the ability of proposed methods to exploit the hierarchy, we lose the ability to use modern representation learning methods that offer a less naive baseline. Moreover, qualitative assessment of the performance is impossible. Ontologue can produce a customized benchmark based on any topic in Wikipedia, and includes node labels and rich data features (Wikipedia abstracts and metadata) to admit appropriate baselines and encourage models that can exploit all available sources of supervision.

**Distribution of labels and data**  With large hierarchies (up to 4000 nodes), the distribution of the labels over the data becomes significantly imbalanced, reducing the problem to an artificially challenging few-shot learning problem in the long tail and an artificially easy multi-label classification problem in the head. Ontologue offers control over the distribution by removing any label that is associated with fewer than a given threshold of data items.

**Hierarchy Structure**  A hierarchy of labels typically models specificity in the underlying universe of discourse: general categories are at the top of the hierarchy and specific categories are at the bottom. As a result, the height and width of the tree can influence the distribution, and therefore performance. Ontologue affords control over the number of hops from the root to offer declarative control over the size, gross graph structure (i.e., height/diameter), and distribution (since distribution correlates with height via specificity).

Also, ontologies are generally DAGs rather than strict hierarchies, and methods that require a hierarchy must pre-process the graph to duplicate children with multiple parents. Although we did not find that this pre-processing step significantly influenced performance, a benchmark should include a significant number of nodes with multiple parents to reflect realistic ontologies. Ontologue assumes the source data and the resulting benchmarks are DAGs to avoid artificially restricting the space.

## 3.2 Declarative Benchmark Generation

We design Ontologue as a simple query interface over DBPedia. [2]. DBPedia [2] is a crowd-sourced community project to extract structured content from several Wikimedia projects[3]. The result is a large multi-domain ontology consisting of 3.77 million "things" with 400 million "facts." The structured information is organized as an Open Knowledge Graph and is publicly available. Variations of DBPedia dataset has served as benchmarks for a variety of machine learning research, such as text classification [46, 50], question answering [22], and information retrieval [41, 36]. DBPedia is under Creative Commons Attribution-ShareAlike 3.0 License and does not include personally identifiable information what would allow doxing or offensive content.

We use the collection from the snapshot for December 2021 [4]. We collect the Wikipedia Article Categories adhering to the Simple Knowledge Organization System (SKOS) schema [21]. The entire graph includes 1,713,451 entities and 4,298,587 edges.

To offer a general impression of the DBPedia graph, we provide a snippet of the top levels of the DBPedia graph. Unlike most knowledge bases which only cover specific domains, DBPedia, as shown in Figure 2, includes a wide variety of domains and represent real community agreement. The ontology carries 3.77 million "things" with 400 million "facts."

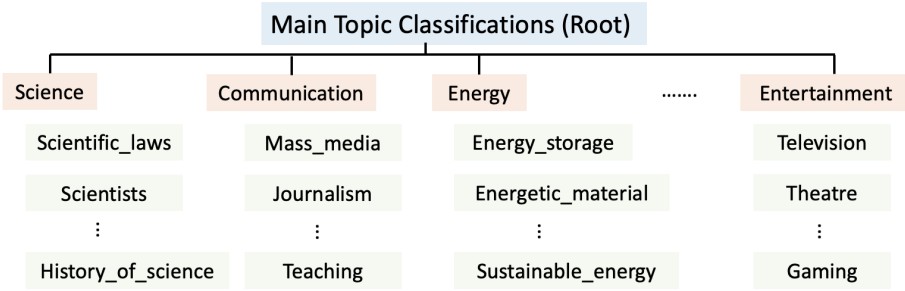

Figure 2: A snippet of the top levels of the DBPedia graph.

**Task**   As shown in Fig. 1, the nature of the task is to classify Wikipedia abstracts into labels which are organized in an ontology structure. The labels are the categories assigned to the main subject of the abstracts. For example, the abstract for "Affidavit" is given labels of "Evidence Law", "Legal Documents", and "Notary".

**Algorithm**   Figure 1 shows the pipeline of Ontologue . Starting from a root concept `TOPIC`, we perform a depth-first search following `skos:broader` relationships between terms to build the label space, and then include all Wikipedia abstracts that use at least one of those labels. Any label that is used in fewer than `MINDATA` data items is ignored, along with its data. The search stops after `MAXHOPS`. The `skos:broader` relationship can include multiple parents for a concept, such that the result may be a directed acyclic graph (DAG).

With these three parameters `TOPIC`, `MINDATA`, and `MAXHOPS`, Ontologue allows benchmark creators declarative, high-level control over context, distribution, and structural properties of the generated datasets, respectively.

**Visual Analysis**   To use Ontologue effectively, the users need to be able to visualize and analyze the results. Ontologue provides two interactive visualizations for evaluating the properties and suitability derived benchmarks: a graph viewer and a distribution viewer. Figure 3 is an illustration of the available visualizations offered by Ontologue . On the left is an interactive graph viewer. The users are provided with options to color code the nodes with given properties and are able to navigate the graph interactively to investigate the context and the structure of the label hierarchy. The interactive graph viewer is created with Pyvis package . The distribution viewer (on the right of the Figure 3) present

---

[3]`https://www.wikimedia.org/`
[4]`https://databus.dbpedia.org/dbpedia/collections/dbpedia-snapshot-2021-12/`

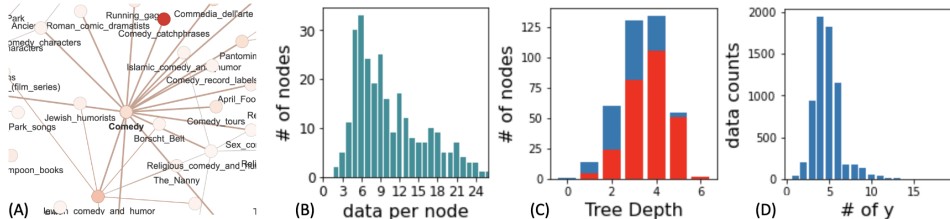

Figure 3: An illustration of the visual analysis features in Ontologue . (A) An interactive graph visualization affording qualitative review of the contextual and structural properties of the derived label hierarchy. (B) A histogram of data distribution over labels. (C) The distribution of easy and hard labels over tree depth. (D) The distribution of the number of labels per data item.

the users various of ways to determine the distributional properties of the generated benchmarks. Ontologue does not provide users total explicit control over a benchmark to avoid cherry picking, but a declarative way to create customized benchmarks.

**Example Benchmarks**  We derived four example benchmark datasets for the topics `Comedy`, `Engineering`, `Law`, and `Main`. `Law` is generated with "Law" as a start node with 10 hops and 30 minimum data points. `Comedy` is created with "Comedy" as the anchor and is allowed 6 hops and minimum 5 data items per class label. To develop `Engineering`, we set maximum hop as 7, minimum data as 6, and "Engineering" as the start node. Finally, `Main` is achieved with 7 hops and 70 minimum data points from the top of the DBPedia graph. The statistics can be seen in Table 1. We will discuss the properties of the derived benchmarks in the next section.

**Generalization**  While we choose DBPedia as our source domains for its popularity in deep learning studies, Ontologue is abstract and easily applied to other sources of datasets as we demonstrate on our GitHub page. We implement Ontologue on MedMentions [35] and provide code for data structure conversion. This property enables users to study their preferred source datasets with ease.

## 4    Performance Results

In this section, we summarize existing benchmarks and four proposed benchmarks derived using Ontologue . We then report on performance results, showing that the proposed benchmarks can change the ranking of competing methods.

### 4.1    Benchmark Statistics

Table 1 shows statistics of the current and proposed benchmarks. We make several observations:

**Size**  We observe that the prior datasets are relatively small in the number of data items, ranging from 1k to 11k. The four proposed datasets, in contrast, range in size from 8k to 61k.

**Domain and Structural Diversity**  Among the prior datasets, there is low diversity in the hierarchies: While the four small datasets cover diverse domains, they are structurally less interesting. There are only two larger, more interesting hierarchies: all *FUNCAT* datasets share the same label hierarchy and all *GO* datasets also share the same label hierarchy. While many papers report on a variety of these "sub" datasets, they are semantically and structurally similar, undermining generalization arguments. The set of four proposed datasets, in contrast, exhibits diversity in size of the data, size of the hierarchy, and the depth of the hierarchy. Moreover, the inputs of all prior datasets are pre-processed feature vectors. Modern methods using learned embeddings provide another source of supervision that can be an appropriate baseline or inform new methods; ignoring this context artificially constrains the solution space.

**Label Distribution: HMC or Few-shot learning?**  The prior datasets are often severely class-imbalanced [17], as shown by the average and median number of data items per label. The long-tailed

Table 1: Statistics of current benchmark and proposed datasets. "# Classes" indicates number of labels in each dataset. "# data" means number of data items. "Averaged #/class" implies average number of data items available per class. "Median #/class" suggests median number of data items available per class. "5-shot" means the percentage of labels with less than 5 data items.

| Taxonomy | Dataset | # Classes | Depth | # data | Averaged #/class | Median #/class | 5-shot |
|---|---|---|---|---|---|---|---|
| DBPEDIA (DAG) | COMEDY | 395 | 6 | 8333 | 80.77 | 13.00 | 20.5% |
| | ENGINEERING | 587 | 7 | 22735 | 226.25 | 31.00 | 14.3% |
| | LAW | 958 | 10 | 61974 | 616.39 | 72.50 | 0% |
| | MAIN | 147 | 7 | 15310 | 529.18 | 139.00 | 0% |
| Tree | ENRON | 56 | 3 | 1648 | 90.2 | 16.0 | 21.4% |
| | DIATOMS | 398 | 3 | 2603 | 10.1 | 5.0 | 48.5% |
| | IMCLEF07A | 96 | 3 | 11006 | 312.5 | 119.5 | 9.4 % |
| | IMCLEF07D | 46 | 3 | 11006 | 652.2 | 137.0 | 19.6% |
| FUNCAT (Tree) | Cellcycle | 499 | 6 | 3757 | 28.5 | 7.0 | 43.7 % |
| | DERSI | 499 | 6 | 3725 | 28.2 | 6.0 | 43.9 % |
| | EISEN | 461 | 6 | 2424 | 21.1 | 5.0 | 46.4 % |
| | EXPR | 499 | 6 | 3778 | 28.6 | 7.0 | 43.7 % |
| | GASCH1 | 499 | 6 | 3764 | 28.5 | 7.0 | 43.7 % |
| | GASCH2 | 499 | 6 | 3779 | 28.6 | 7.0 | 43.7 % |
| | SEQ | 499 | 6 | 3919 | 29.1 | 7.0 | 43.5 % |
| | SPO | 499 | 6 | 3703 | 28.0 | 6.0 | 44.3 % |
| Gene Ontology | Cellcycle | 4122 | 12 | 3751 | 12.5 | 2.0 | 71.7 % |
| | DERSI | 4116 | 12 | 3719 | 12.4 | 2.0 | 71.8 % |
| | EISEN | 3570 | 12 | 2418 | 10.3 | 1.0 | 72.9 % |
| | EXPR | 4128 | 12 | 3773 | 12.5 | 2.0 | 71.7 % |
| | GASCH1 | 4122 | 12 | 3758 | 12.5 | 2.0 | 71.6 % |
| | GASCH2 | 4128 | 12 | 3773 | 12.5 | 2.0 | 71.7 % |
| | SEQ | 4130 | 12 | 3900 | 12.9 | 2.0 | 71.4 % |
| | SPO | 4116 | 12 | 3697 | 12.3 | 2.0 | 71.9 % |

distribution of the number of data items per label suggest that the challenge of these datasets is not in the hierarchical structure, but in the embodiment of a conventional few-shot learning problem.

To further investigate this issue, we calculated the percentage of nodes with lower than 5 pieces of data points relative to the whole hierarchy and Table 1 shows the result. For reference, current benchmarks for few-shot classification are MiniImagenet [43], CIFAR-FS [3], and CUB [44]. The MiniImagenet, a sub set of the Imagenet[24], includes 100 classes and 600 images per class. The standard procedure is to split into 64 classes for training, 16 novel classes for validation and 20 novel classes for testing. The CIFAR-FS dataset also has 100 classes with 100 images per class. The set up for few shot learning is the same as the MiniImagenet. The CUB dataset has 200 classes with 11,788 images in total (averaged 58.9 images per class). The dataset is split into 100 base classes, 50 novel classes for validation, and 50 novel classes for testing [19]. The percentage of novel classes to all classes in the test set for MiniImagenet, CIFAR-FS, and CUB are 20%, 20%, and 25%, respectively.

Some of these HMC benchmark datasets are arguably more difficult than standard few-shot benchmark datasets: (i) There are a higher number of few-shot classes in *DIATOMS*, the *FUNCAT* collection, and the *GO* collection. (ii) There is less data available per class for training in all but 1 HMC datasets than few-shot datasets (iii) Few-shot learning algorithms often rely on alternative sources of supervision or significant pre-training to predict novel classes. These resources are not accessible in HMC benchmarks due to the limited information of the datasets. While it is practical that limited data is available for large amount of labels in the real-life HMC problems, the few-shot aspect of HMC problems has never been explored in existing HMC studies. We believe a clear characterization of the HMC benchmark datasets will provide a novel direction and facilitate HMC research.

## 4.2 Evaluation of State of the Art Methods

We report performance of a naive baseline and several state of the art methods against the four proposed benchmarks.

**Naive Baseline** is a three-layered feed forward network optimized with binary cross entropy. This network is similar to basic multi-label classification model. **Surj** is the first HMC algorithm integrating Graph Neural Network in its framework. Surj learns a representation of the label hierarchy with a graph autoencoder and maps the input onto the representation space with cosine similarity. The network is also optimized with binary cross entropy. **C-HCMNN**[5] [17] is designed to produce coherent predictions which respect hierarchical structure. The network adopts a modified binary cross entropy loss that enforcing the predicted probabilities of children nodes are lower than their ancestors. **HMC-GA**[6] [10] is a global method too induce HMC rules which are generated using a deterministic procedure considering the classes assigned to instances. **CLUS-HMC**[7] [42] is based on Predictive Clustering Trees (PCT) to generate a decision tree to cover the entire tree hierarchy. **CLUS-HMC-ENS** [39] significantly improves upon CLUS-HMC by using a bagging strategy to create ensembles of Clus-HMC trees.

**AUPRC**  We use standard metric for HMC $AU(\overline{PRC})$ to evaluate the performance of baseline methods on the four benchmarks. $AU(\overline{PRC})$ is universally adopted [45, 49, 17] to evaluate the performance of HMC algorithms. $AU(\overline{PRC})$ is defined as the area under the average precision and recall curve. Utilizing $AU(\overline{PRC})$ allows studies to avoid thresholding, in which the selection of the threshold can be application-dependent, arbitrary and difficult to acquire.

**Global Hierarchy Violations**  We also measure the compliance with hierarchy constraints using the Global Hierarchy Violations (GHV) measure introduced in [49]. HMC methods assign a probability to each label. A violation occurs when an ancestor is assigned a lower probability than its descendant. GHV is the count of all such violations.

Many HMC models are designed to avoid hierarchy violations. Wehrmann et al. [45] use a regularizer to penalize hierarchical violations and Giunchiglia et al. [17] employ a modified binary cross-entropy loss (MCLoss), which constrains the predicted probability of a descendant node to only be as high as its ancestors. While both algorithms improved upon the state of the art in $AU(\overline{PRC})$, both studies failed to demonstrate whether any hierarchy violations actually occurred. Global Hierarchical Violations was proposed as a measure to address this issue [49].

**Implementation**  For each dataset, we run all algorithms 10 times and calculate the averaged $AU(\overline{PRC})$ and the normalized Global Hierarchical Violation. The normalized Global Hierarchical Violation computes the Global Hierarchical Violations over all valid node pairs. All algorithms do not consider auxiliary information (such as text label for each node) of the label hierarchy for fairness. Ontologue and the four proposed datasets will be available on GitHub[8].

Table 2 shows the results of all methods on the four new proposed benchmarks. We see that unlike prior results [49], which showed that Surj was superior on all but three datasets, there is more diversity in the ranking. We conclude that these benchmarks can reveal limitations to generalization and inform research in new classes of methods. We note, however, that when Surj is free to use the node labels as a source of additional supervision (Table 3), it remains the state of the art.

## 5 Comparative Analysis of Benchmarks

In this section we perform additional experiments to study the quantitative impact of improving benchmark diversity. We consider requirements around context, label distribution, and structure.

**Application Context**  Because our benchmarks provide context, e.g. text associated with the labels, methods can take advantage of this additional information. Surj framework is able to consider label

---

[5]`https://github.com/EGiunchiglia/C-HMCNN`

[6]`http://www.biomal.ufscar.br/hmc.html`

[7]`https://dtai.cs.kuleuven.be/clus/`

[8]`https://github.com/seanyang38/Ontologue`

Table 2: $AU\overline{PRC}$ and normalized Global Hierarchy Violations (parentheses) on proposed benchmarks for several state of the art methods and a naive baseline based on binary cross entropy loss. The naive baseline outperforms all but the top two methods, suggesting that some previous results were confounded by label distribution issues.

|  | COMEDY | ENGINEERING | LAW | MAIN |
|---|---|---|---|---|
| Naive | 0.839 (0.013) | 0.829 (0.007) | 0.777 (0.019) | 0.782 (0.038) |
| Surj | 0.857 (0.005) | **0.853 (0.003)** | **0.796 (0.008)** | 0.809 (0.006) |
| C-HMCNN | **0.870 (0.004)** | 0.842 (0.005) | 0.783 (0.010) | **0.819 (0.006)** |
| HMC-LMLP | 0.627 (0.012) | 0.730 (0.016) | 0.671 (0.019) | 0.663 (0.018) |
| CLUS-HMC | 0.620 (0.023) | 0.603 (0.035) | 0.632 (0.025) | 0.640 (0.020) |
| CLUS-ENS | 0.713 (0.014) | 0.695 (0.027) | 0.693 (0.013) | 0.734 (0.013) |

Table 3: Surj performance ($AU\overline{PRC}$ (GHV)) with and without text labels. Performance improves on all datasets when text labels are available, making Surj the state of the art since other methods cannot make use of the text.

|  | COMEDY | ENGINEERING | LAW | MAIN |
|---|---|---|---|---|
| Surj w/ label text | 0.872 (0.004) | 0.873 (0.002) | 0.802 (0.004) | 0.834 (0.005) |
| Surj w/o label text | 0.857 (0.005) | 0.853 (0.003) | 0.796 (0.008) | 0.809 (0.006) |

features. We analyze whether this contextual information improves the performance of Surj on the proposed benchmarks. We embed label text with BERT[9] and serve the embeddings as node features during graph auto-encoder training. We run the rest of the framework with the same procedure. Table 3 shows the performance on Surj with and without considering label text. Surj improves on all datasets for both $AU(\overline{PRC})$ and normalized Global Hierarchical Violation. It implies that contextual information from the graph can help HMC problem.

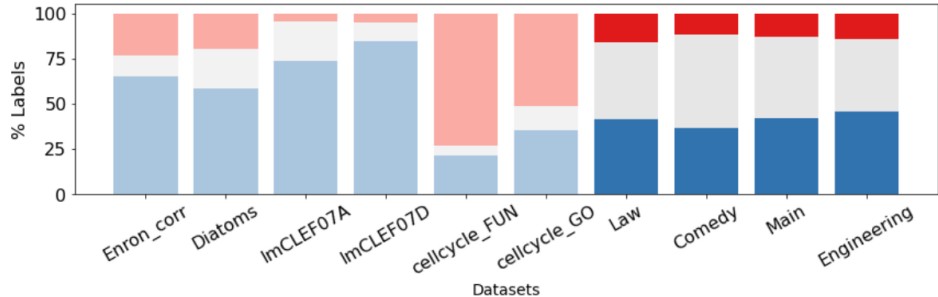

Figure 4: The percentage of "trivial" labels for which all methods are successful (blue), "impossible" labels for which no method is successful (red), and all other labels (gray). Ontologue benchmarks have a significantly lower proportion of trivial and impossible labels (45% to 58%), and are therefore more useful for analyzing performance and generalizability.

**Distribution**  Meding et al. [32] find that ImageNet validation set suffers from dichotomous data difficulty (DDD): 57.5 % of the images are either "trivial" (too easy) or "impossible" for the model to recognize. We adopt similar methodology used in the study to analyze the difficulty of the current and proposed benchmarks. Instead of looking at data items, we consider the difficulty of the labels. We compute $AU(\overline{PRC})$ for each label for all 6 baseline algorithms. The label is regarded as "trivial" when 5 or more models have over 0.5 $AU(\overline{PRC})$. On the other hand, the label is considered as "impossible" when 5 or more models have less than 0.5 $AU(\overline{PRC})$. We calculate the percentage of "trivial" and "impossible" labels from all datasets and plot the results in Figure 4. Red/Pink indicates "impossible" examples, and blue means "trivial" labels, and gray shows interesting cases where the

---

[9]We use the pre-trained model "all-MiniLM-L6-v2" from `https://github.com/UKPLab/sentence-transformers`

current methods have a variety of performance. The current benchmarks have over 75 % of labels are either too easy or impossible for the models to solve, while the proposed benchmarks range from 45 % to 58 %. This analysis implies that the proposed benchmarks provide diversity and create confusion between models.

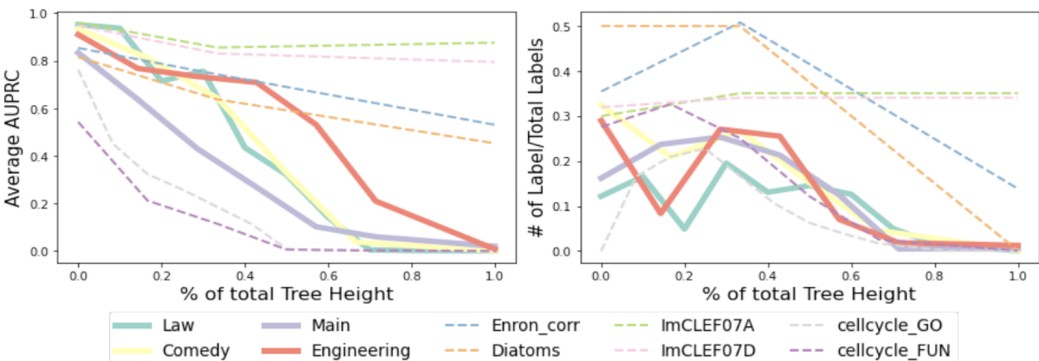

Figure 5: Left: Average $AU(\overline{PRC})$ over tree depth. Performance is influenced by tree height. Right: Distribution of labels over tree depth. Performance of GO and FUNCAT benchmarks are dominated by low-coverage labels at the bottom, while other benchmarks are shallow and therefore unrealistically independent of tree structure. Ontologue datasets (in bold) offer a smooth decline over hierarchy depth (left) and a more balanced distribution (right).

**Structure** Finally, we analyze the structural property of the proposed benchmarks. The left chart of Figure 5 shows the average $AU(\overline{PRC})$ by tree depth and the right chat demonstrate the distribution of labels over tree depth. The experimental results are average of naive baseline, Surj, and C-HMCNN. We can observe in the left chart that the current benchmarks are extreme: the depth of the trees has either limited impact (four dashed lines on the top) or significant impact (two dashed lines at the bottom) to the performance. The proposed benchmarks (solid thicker lines) demonstrate a more smooth decline over tree depth and this presents a more reasonable problem space for HMC.

# 6 Discussion and Conclusion

**Bias and Limitation** While DBPedia and WikiPedia are prone to Neutral Point of View (NPOV), they still suffer from bias and limitations from its crowd-sourced nature. Many studies have shown bias in gender, culture, and political ideology[20, 6, 30]. It is also known that Wikipedia does not include consistently accurate and sufficient information in high-expertise domains [48, 26]. Yacob et al. [48] showed that Wikipedia presented more errors of omission compared to Schwartz Principles of Surgery, a surgical textbook and Lavsa et al. [26] introduced the finding that knowledge on contraindications and precautions, drug absorption, and adverse drug events was often found to be wrong. It is important to be aware of these inherent weaknesses from DBPedia and WikiPedia when using Ontologue, especially for high-expertise applications. In addition, our datasets are based on a single (though diverse) source of text, and the declarative interface could afford cherry-picking designs through trial and error. Further, the Wikimedia data also carries a small amount of risk to those whose personal information is represented there.

In this paper, we demonstrate that current HMC benchmarks exhibit degenerate statistical and structural properties that can confound performance analysis among competing methods. We present a declarative toolkit, Ontologue , for generating custom, realistic, and diverse benchmarks that can better assess claims of generalizable performance. We use Ontologue to derive four new benchmarks using DBPedia and show that these benchmarks provide application context that can afford multi-modal approaches, more representative distribution of data over labels, and more realistic structural properties. We conclude that these benchmarks and the Ontologue system itself will improve the diversity of representative tasks and continue to lead to new insights in the analysis of graph-oriented labeling tasks.

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
