# 7 Supplementary Materials

We report additional data and experimental results. We elaborate Global Hierarchical Violation in Section 7.1, offer a snippet of the DBPedia graph in Section **??**, and provide supplementary comparative analysis of benchmarks in Section 7.2. Checklist is provided in the last page.

## 7.1 Global Hierarchy Violation

We elaborate Global Hierarchy Violation with illustrations. Consider the scenario in Figure 1(b), GHV calculates the number of the violations in 8 node pairs: A-B, A-D, A-E, A-C, A-F, B-D, B-E, and C-F. Among these pairs, A-F and C-F do not follow hierarchy constraint, so the GHV would be 2 in this scenario. While GHV sufficiently measure hierarchy violations, Yang and Howe also acknowledge that GHV has a blind spot. Figure 1 (c) presents a setup in which the pattern of the predicted probabilities is similar as presented in 1 (B), but the predicted probabilities in 1 (c) are much lower. Practically, thresholds are usually set around 50% to acquire labels. Hierarchy violations occur in this scenario would be irrelevant after thresholding.

The main focus of HMC problems is to produce coherent predictions, i.e. the predicted labels respect hierarchy constraint. Hierarchy violation occurs when child nodes are predicted without parent nodes, as shown in Figure 1(a). For predicted probabilities, hierarchy violation happens when $p_B(x) > p_A(x)$ where $p(x)$ indicates predicted probability of input $x$ and $B$ is a child node of $A$. A demonstration is shown in Figure 1(b).

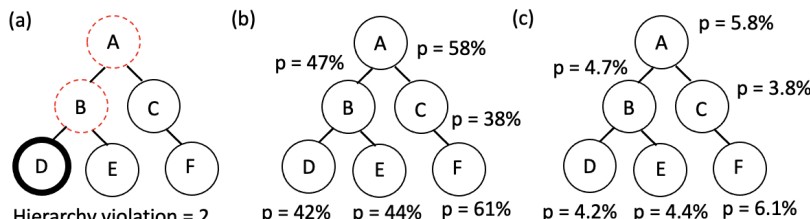

Figure 1: Illustration of hierarchy violation with (a) predicted labels and (b) predicted probabilities (c) low predicted probabilities. Letters identify nodes. Bold line circles predicted labels and red dashed lines indicates labels with hierarchy violations (a) Hierarchy violations occur in this scenario because a descedant node D is predicted without its ancestors nodes A and B (b) In this case, $p$ annotations indicate predicted probabilities. Hierarchy violations occur when the predicted probability of a child node is higher than that of one of its ancestors. A-F and A-C pairs are hierarchy violations. (c) The predicted outputs share the same patter as (b) with much lower values. Thresholds in real life applications are often set around 50%. Although hierarchy violations are presented in this scenarios, these violations would be irrelevant after thresholding.

## 7.2 Additional Comparative Analysis of Benchmarks

We demonstrate supplementary comparative analysis of the current and Ontologue benchmarks.

**Distributional Analysis** We perform four data distributional analysis on current benchmarks (Figure 3) and proposed benchmarks (Figure 2): (i) Data availability for labels (first column) (ii) Data distribution across hierarchy levels (second column) (iii) Node and leaf (marked in red) distribution across hierarchy levels with (third column) (iv) Label distribution over data (forth column).

- **Data availability for labels (first column)**: Each chart shows a histogram describing data availability for each node for the corresponding dataset. The histograms confirm that HMC datasets are often class imbalanced [**?** ]. Significant number of nodes in *Diatoms*, *FUN* collection, and *GO* collection have less than 5 data points to train on. The long-tailed distribution of the number of data items per label suggest that the challenge of these datasets is not in the hierarchical

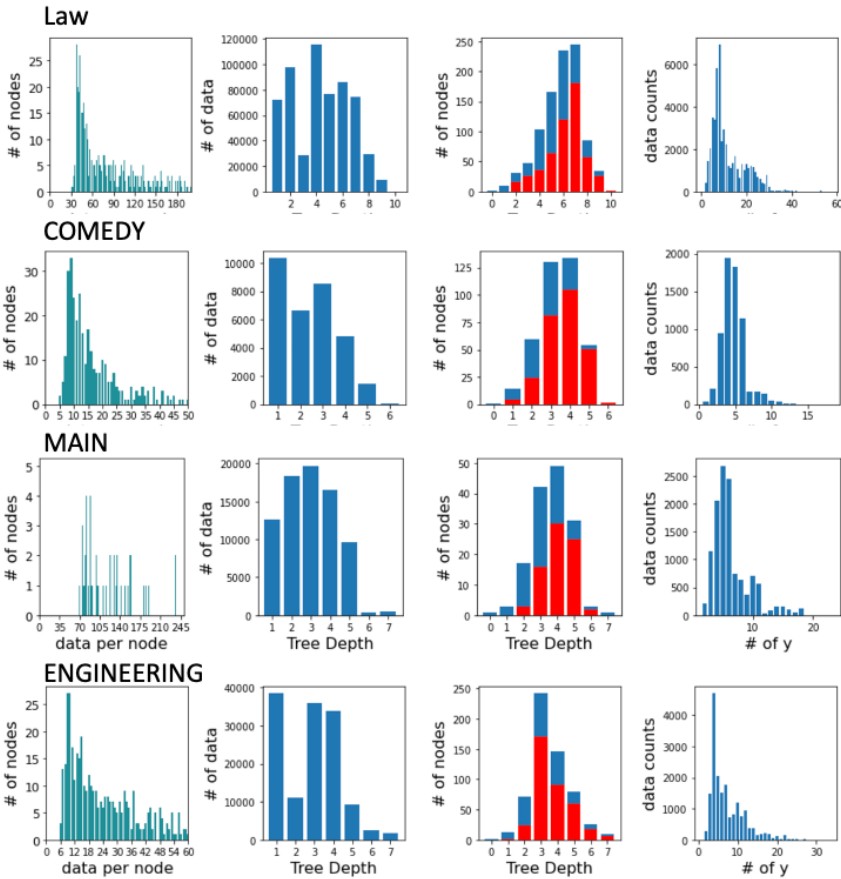

Figure 2: Distribution analysis for the proposed benchmarks. The first column is the data availability for labels. The second column shows the data distribution over the label hierarchy levels. The third column demonstrates the node distribution over hierarchy levels with red bars indicate leaf (nodes without children) distribution over hierarchy levels. The forth column illustrates label distribution over data.

structure, but in the embodiment of a conventional few-shot learning problem. Our proposed toolkit and benchmarks address this issue by setting minimum data availability for the labels. This provide proper benchmarks for HMC algorithms to reduce the influence of the few-shot problems and evaluate the performance over hierarchical structure.

- **Data distribution across hierarchy levels (second column)**: Nodes at deeper levels often have lower data availability compared to nodes at higher levels as we can observe in *DIATOMS*, *FUNCAT*, *GO* collections and in proposed benchmarks. Our proposed benchmarks still carries high data availability at the deep levels.

- **Node and leaf distribution across hierarchy levels (third column)**: The label hierarchies of *ENRON*, *DIATOMS*, *IMCLEF* are shallow and the leaves (nodes without children) are all located in the bottom two levels. This undermines the difficulty of HMC problem provided from the label hierarchy. The proposed benchmarks have more organically constructed label hierarchies with leaves range from level 2 to level 10.

- **Label distribution over data**: We discover that three of the datasets, *DIATOMS*, *ImCLEF07A*, and *ImCLEF07D* are not necessarily multi-label datasets because every data point is only assigned to one branch of labels. This significantly reduces the difficulty of the problem.

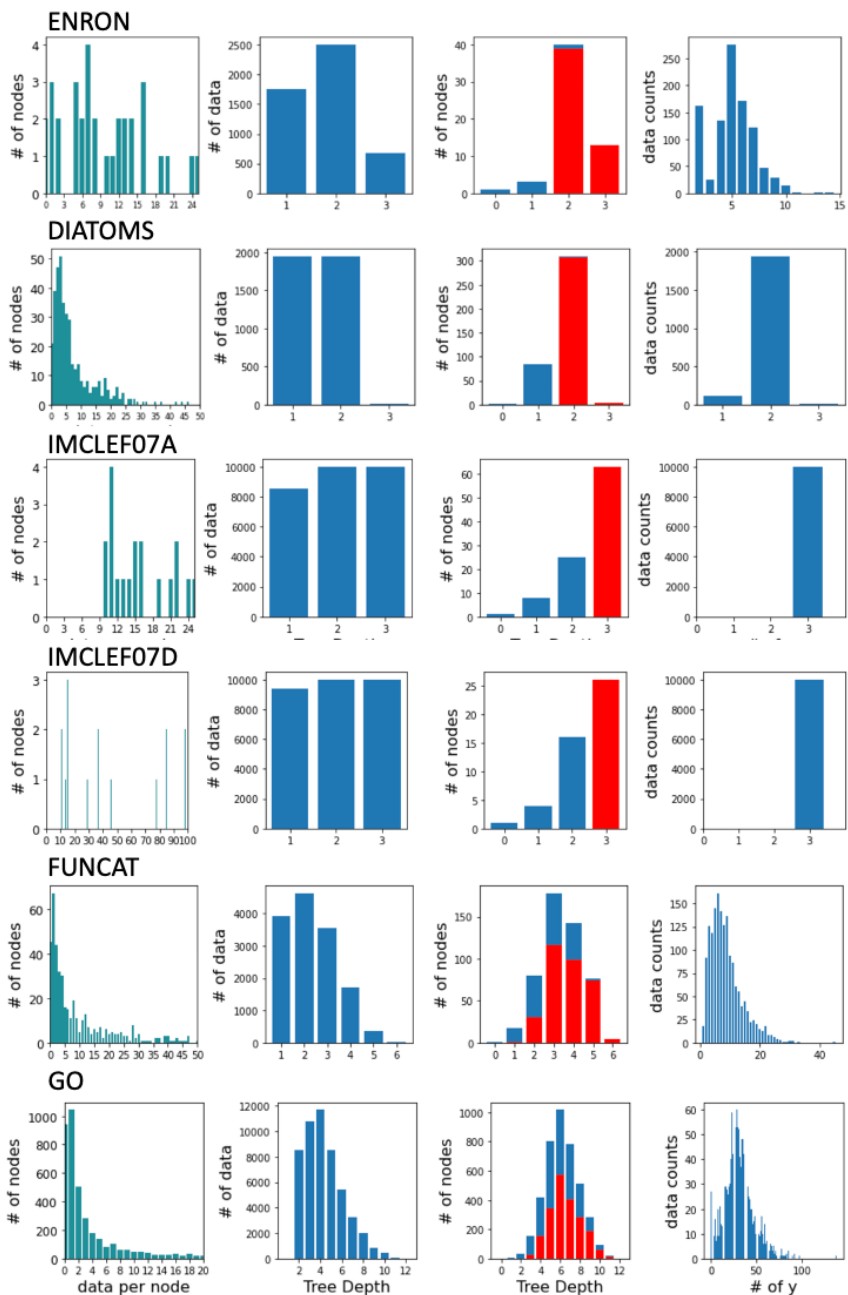

Figure 3: Distribution analysis for the current benchmarks. The first column is the data availability for labels. The second column shows the data distribution over the label hierarchy levels. The third column demonstrates the node distribution over hierarchy levels with red bars indicate leaf (nodes without children) distribution over hierarchy levels. The forth column illustrates label distribution over data.

The current benchmarks can be seen as tri-modal: There are three small and shallow hierarchies (*ENRON*, *DIATOMS*, and *IMCLEF*) from diverse domains with less than 100 nodes, a related set of functional genomic hierarchies of moderate size (*FUNCAT*), and gene ontology hierarchies where the number of labels is approximately the same as the number of data items *GO*. Since the size and structure correlates with domain, it becomes difficult to determine how performance results may

Table 1: Average number of parents (not considering roots) and children (not considering leaves) of the current and proposed benchmarks. Label hierarchies in most current benchmarks are trees, so the number of the parents for all node is 1. Only three nodes in the *ENRON* label hierarchy are internal nodes (not a root or leaf) and they carry a lot of children nodes. The label hierarchies in the Ontologue datasets are more diverse and complex and provide a more difficult HMC problem space.

| Taxonomy | Dataset | Averaged # of parents | Averaged # of children |
|---|---|---|---|
| DBPEDIA (DAG) | COMEDY | 1.33 | 2.96 |
| | ENGINEERING | 1.14 | 3.52 |
| | LAW | 1.29 | 2.68 |
| | MAIN | 1.52 | 4.12 |
| Tree | ENRON | 1.00 | 15.00 |
| | DIATOMS | 1.00 | 3.12 |
| | IMCLEF07A | 1.00 | 2.66 |
| | IMCLEF07D | 1.00 | 2.11 |
| | FUNCAT | 1.00 | 2.71 |
| DAG | GO | 1.41 | 2.78 |

generalize — we do not have large hierarchies in social domains or small hierarchies in biological domains, for example. Ontologue datasets provide diverse data distributions and tree structure to improve the generalization of the HMC benchmarks.

**Structural Analysis**    We investigate the structure of the label hierarchies by computing the averaged number of children and parents. Table 1 demonstrates the comparison between current benchmarks and Ontologue datasets. We disregard the root node when we calculate the average number of parents and ignore the leaves when we compute the average number of children. Label hierarchies in most current benchmarks are trees, so the number of parents for all nodes is 1. *Enron* is an extreme case given that there are only three internal nodes, not a root or leaf, in the label hierarchy. We can observe that the Ontologue datasets carry more complex and diverse hierarchies. The complexity of the label hierarchies provides a more difficult problem space for HMC.