# OpenReview forum: "Ontologue: Declarative Benchmark Construction for Ontological Multi-Label Classification"
_NeurIPS.cc/2022/Track/Datasets_and_Benchmarks — NeurIPS 2022 Datasets and Benchmarks _

### Official Review · Reviewer_PZRo · 2022-07-20
**In this paper, the authors analyze the weaknesses of multiple HMC benchmark datasets and a toolkit Ontologue is designed to construct benchmarks for HMC. The analysis is comprehensive and this toolkit is beneficial to evaluate the performance of HMC methods correctly.**

**Rating:** 6
**Confidence:** 4
**Clarity:** This paper is well written.

**Strengths:**

1. This toolkit takes the application context, distribution of labels and data, and hierarchy structure into consideration.
2. It can be used to produce a customized benchmark based on any topic in Wikipedia, which contributes to the accurate evaluation of the effectiveness of the HMC methods.

**Weaknesses:**

1. The figures in this paper are not clear enough. Please modify them.
2. This proposed system called Ontologue is not open so far, and the proposed datasets in the GitHub repository are empty.

**Additional Feedback:**

1. Can the authors give an example about the use of the toolkit? What are the input and output of the toolkit? Please explain the whole generation process step by step.
2. In the right part of Figure 3, what do the X-axis and the Y-axis mean, respectively? How to calculate the distribution of labels in this figure? What is the purpose of demonstrating the distribution of labels over tree depth?
3. The terminology “DAG” is first used without its full name in line 69.

**Correctness:**

The claims made in the submission are correct. The datasets proposed in this paper are constructed soundly.

**Documentation:**

Yes. There is sufficient detail on data collection and organization, availability and maintenance, and ethical and responsible use. But there is no documentation related to hosting, licensing, and maintenance plan. The proposed system and datasets are not accessible so far.

**Ethics:**

No, there isn’t.

**Relation To Prior Work:**

Yes, this paper clearly discusses the existing benchmarks' weaknesses, and comprehensive analysis of the new benchmarks against the old ones is presented.

**Summary And Contributions:**

1. This paper analyzes the weaknesses of the existing 20 HMC benchmark datasets.
2. This paper proposes a toolkit called Ontologue to construct customized benchmarks.
3. The authors derive four new benchmarks and evaluate the performance of state-of-the-art models.
4. The authors perform a comparative analysis of the new benchmarks against the old benchmarks.

---

> ### Author Response · Authors · 2022-08-23
> **Response to Reviewer PZRo**
>
> We appreciate the comments.  (All mentions of the paper content indicate the first submission unless specified)
>
> **The figures in this paper are not clear enough. Please modify them.**
>
> We thank the feedback on the clarity of our figures. We would appreciate a little more specifics on what needs to be improved.
>
> **This proposed system called Ontologue is not open so far, and the proposed datasets in the GitHub repository are empty.**
> We apologize for the delay. The code and documentation have been updated.
>
> **Can the authors give an example about the use of the toolkit? What are the input and output of the toolkit? Please explain the whole generation process step by step.**
>
> We believe the structure of the label hierarchy and the data coverage over the labels are the key elements of the HMC problem. How the hierarchy structure and the data coverage influence the performance is still an open question. Ontologue will allow the researchers to control over the maximum depth of the hierarchy and the minimum data records required for each label. With these functions, we will be able to study specific aspects of the HMC problem.
> To provide more clarity, we add a figure to describe the pipeline of Ontologue (Figure 1 in the revision). Here is an example of how an user would interact with Ontolgue: (1) The input is set to be DBPedia data by default. It includes the DBPedia ontology, short abstracts on Wikipedia and the labels for these abstracts on the ontology. Ontologue is also easily applied to a different source. The user will need to provide a set of data records, the labels, and the ontology of the labels. (2) Then, the user provides the preferred starting node, the maximum hop, and the minimum data records for each label. (3) Ontologue will generate a dataset based on these requirements. The statistics and the visualizations will also be created to help the user understand the generated dataset.
>
> **In the right part of Figure 3, what do the X-axis and the Y-axis mean, respectively? How to calculate the distribution of labels in this figure? What is the purpose of demonstrating the distribution of labels over tree depth?**
>
> These two figures are meant to reveal the structural patterns of the benchmarks . The x-axis for both figures is the normalized tree depth. The y-axis for the figure on the right is the percentage of the label in each level. It is computed as the number of the labels in the level divided by the total number of the labels. The chart gives us an overall idea how the shapes (top heavy, skinny, average) of the label hierarchies compare to each other. It shows that the labels from the proposed datasets are evenly distributed until the last couple levels.
>
> **The terminology “DAG” is first used without its full name in line 69.**
>
> Thank you for pointing this out. We include the full term, Directed Acyclic Graph (DAG), in the revision (line 69).

---

> > ### Comment · Reviewer_PZRo · 2022-08-29
> > **I am satisfied with the author's responses.**
> >
> > In the revised version, the authors have presented their work more clearly, and I am satisfied with that. Thank you.

---

### Official Review · Reviewer_ZeEZ · 2022-07-26
**First version of an official review of the paper "Ontologue: Declarative Benchmark Construction for Ontological Multi-Label Classification"**

**Rating:** 8
**Confidence:** 4
**Correctness:** The experiments and the datsets are d…
**Clarity:** The paper is well-written. It is easy…

**Strengths:**

1. HMC is an important problem. This paper increases our understanding of HMC methods by inroducing data for method benchmarking;
2. The Ontologue tool allows to extract such datasets from DBPedia, which meet experimental needs. At the same time the Ontologue tool prohbits cherry picking behaviour;
3. The four benchmarks create a reasonable problem space for HMC methods.

**Weaknesses:**

1. [minor] Along with four datasets in English, the authors could extract one or two datasets in other languages;
2. [minor] At the moment the GitHub repo of the paper is empty. The authors are requested to update the repo.
3. [minor] Line 283: For reproducability reasons the authors are requested to state explicitly which BERT model was used.

**Additional Feedback:**

Please consider a few minor issues, described above.

**Documentation:**

The code for reproducing the experiments is not available.

**Ethics:**

No ethical issues.

**Relation To Prior Work:**

The paper is grounded in previous works on HMC.

**Summary And Contributions:**

1. The authors develop a toolkit Ontologue, which allows to extract HMC datasets from DBedia, which meet certain pre-defined conditions;
2. The authors extract four datasets and evaluate performance of SoTA methods using them;
3. The authors have conducted an extensive analysis of existing HMC benchmarks and revealed their limitations and challenges;
4. The paper presents several novel and useful analyses of the HMC datasets with respect to descriptive statistics, domain and structural diversity,  label distirbution and Global Hierarchy Violations.

---

> ### Author Response · Authors · 2022-08-23
> **Response to Reviewer ZeEZ**
>
> **Along with four datasets in English, the authors could extract one or two datasets in other languages**
>
> We thank you for the comments. We appreciate the suggestion to extract datasets in other languages. Ontologue is easily applied to other sources of datasets, as we show on Github. We encourage researchers to apply Ontologue to their preferred languages from DBPedia.
>
> **At the moment the GitHub repo of the paper is empty. The authors are requested to update the repo.**
>
> We apologize for the delay. The code and documentation have been updated.
>
> **For reproducibility reasons the authors are requested to state explicitly which BERT model was used.**
>
> Thank you for pointing this out. We use the pre-trained model "all-MiniLM-L6-v2" from https://github.com/UKPLab/sentence-transformers. We include the version of the BERT model in our revision (line 292).

---

> > ### Comment · Reviewer_ZeEZ · 2022-08-31
> > **No changes in score**
> >
> > After carefully reading other reviews and the author's responses, this reviewer retains a score of 8 (clear accept).

---

### Official Review · Reviewer_tU6w · 2022-07-28
**Framework for generating HMC datasets for specific kinds of models.**

**Rating:** 7
**Confidence:** 3
**Clarity:** Paper is well written.

**Strengths:**

- The Ontologue framework seems useful for furthering methods for HMC, it is also nice that it comes with a framework to visualize the generated datasets.
- The experiments presented demonstrate that the datasets generated with Ontologue offer a meaningful challenge to models (Fig 2).


**Weaknesses:**

- The motivation of the paper seems to be based too much on a specific model for HMC developed by the authors in prior work (Surj). The paper uses Surj to demonstrate that specific desirable properties are fulfilled by the dataset (eg the benefit of text associated with labels - lines 280-287) but also argues that these properties are important for leveraging modern few shot learning methods (lines 50-55). This seems like a circular argument which makes it seem like the datasets built here were built to demonstrate the capabilities of a model with focus on a realistic task somewhat lost. I would encourage broader discussion and framing of the challenges presented by the dataset beyond those specific to one model.

- The paper also repeatedly (lines 79-81, line 157) argues that the proposed framework generates datasets which are more "realistic" than existing benchmarks. Given that the existing datasets are realistic datasets and the proposed allows one to generate synthetic datasets this claim is strange. What makes a dataset "realistic"? For example, what makes nodes with multiple parents "more realistic"? I would encourage greater discussion of what makes the generated datasets more realistic (quantitative comparisons to other hierarchical datasets (eg perhaps ones not used for HMC?) and perhaps more grounding in the literature) or discuss the implications of the proposed framework which generates challenging synthetic (yet challenging) datasets -- synthetic + challenging wouldn't be a slight on this work.

- The exact nature of the task proposed in the paper is unclear. What exactly is the input "abstract" and what do the labels associated with it say about the input?

**Additional Feedback:**

Line 273: "We see that unlike prior results, which showed that Surj was superior on all but three datasets, there is more diversity in the ranking." -- what are the prior results? Please summarize if this is referencing results from other papers.

Figure 3: Why is performance influenced by tree height?

How does the proposed dataset connect/differ from entity linking datasets such as [MedMentions](https://github.com/chanzuckerberg/MedMentions) or [TypeNet](https://aclanthology.org/P18-1010.pdf)?

**Post author response:**

Dear authors, thank you for your responses. I think these are reasonable and have updated my score. I would also encourage discussion of entity linking datasets in Sec 2 perhaps, especially since you also run Ontologue on MedMentions.

**Correctness:**

The claims seem somewhat correct (discussed in weakness above). Additional questions:

1. Given that other datasets for HMC are not text datasets (eg protein function prediction or medical image annotation), what is the extent to which models developed with Ontologue will be applicable on these other datasets? Eg would it be possible to obtain meaningful text associated with labels in these datasets? I would encourage discussion of the framework to other application domains with realistic HMC problems.

2. "We intentionally designed Ontologue to limit explicit, low-level control over benchmark properties to discourage the creation of cherry-picked datasets that would invalidate generalizability claims." - what exactly are these design elements? What is an example of cherry picking in the case of Ontologue which is prevented by its design? What are the benchmark properties?


**Documentation:**

The github repository associated with Ontologue seems empty: https://github.com/seanyang38/Ontologue

**Relation To Prior Work:**

Relation to prior work is mostly clear. Exceptions:

1. "Some of these HMC benchmark datasets are arguably more difficult than standard few-shot benchmark datasets" -- I would highly encourage the discussion of specific existing few-shot benchmark datasets in this paragraph.

**Summary And Contributions:**

The main proposal in this paper is a way for filtering the dbpedia dataset so that it can be used for a hierarchical classification task, called Ontologue. Ontologue allows control over the depth of the tree on the label space, the number of examples in the specific label, and the topic of the dataset based on the choice of the starting node. The paper shows the usefulness of Ontologue by generating 4 different datasets and running baseline and state of the art HMC models on the datasets. The paper also presents the argument that existing HMC datasets present 2 artificial problems, with some examples having a multi-label classification and another sizable set of examples having a few shot learning problem. They also note that existing datasets dont text data associated with labels which may be leveraged by modern few shot learning methods. The datasets which one can generate with Ontologue remedy this.

---

> ### Author Response · Authors · 2022-08-23
> **Response to Reviewer tU6w (1/2)**
>
> We appreciate the extensive consideration and comments.  (All mentions of the paper content indicate the first submission unless specified)
>
> **The motivation of the paper seems to be based too much on a specific model for HMC developed by the authors in prior work (Surj)....**
>
> We thank you for your comment and we understand your consideration. We established notable and concerning weaknesses (Section 4.1) in the current HMC benchmarks, which include lack of structural diversity and extreme low data coverage.
>
> We show that “data item features may be pre-processed using outdated feature engineering techniques” (line 136) and provide evidences “ImCLEF07A and ImCLEF07D include Local Binary Pattern (LBP) and Edge Histogram Descriptors (EHD) for texture features and Scale-Invariant Feature Transform (SIFT) for local features of X-Ray images” (line 117-119) and  “Diatoms is a dataset to classify images of diatoms into hierarchical taxonomic rank.  The feature attributes are products of Fourier descriptors and SIFT descriptors.” (line 121-122) These more conventional feature engineering do not reflect modern computer vision domains.
>
> Additionally, we point out that (line 218-227) the current HMC benchmarks are arguably more difficult than the few-shot benchmarks and these aspects of the problem have been ignored by the existing studies.
>
> We believe these flaws in current HMC benchmarks have created a bottleneck in the HMC problem. Ontologue and the proposed datasets are designed to tackle these weaknesses. Surj, our prior work, does have properties that can take advantage of the context of the label hierarchy, but that should not shadow our analysis and contributions.
>
> **Given that the existing datasets are realistic datasets and the proposed allows one to generate synthetic datasets this claim is strange…..I would encourage greater discussion of what makes the generated datasets more realistic..**
>
> We appreciate your concern. First, we disagree that the proposed datasets are “synthetic”. The proposed datasets are the subsets of the whole DBPedia dataset. None of the content of the data was computer-generated. Second, as we mention in Section 3.1 “node labels have been replaced with integers, and data item features may be pre-processed using outdated feature engineering techniques”, the current benchmarks have been artificially stripped of useful information and that the feature engineering of the current datasets are outdated. For example, in line 117-119, “ImCLEF07A and ImCLEF07D include Local Binary Pattern (LBP) and Edge Histogram Descriptors (EHD) for texture features and Scale-Invariant Feature Transform (SIFT) for local features of X-Ray images” and line 121-122 “Diatoms is a dataset to classify images of diatoms into hierarchical taxonomic rank.  The feature attributes are products of Fourier descriptors and SIFT descriptors.” These features are not commonly used to solve current computer vision problems. We do not believe these are realistic datasets to serve as modern HMC benchmarks.
>
>
> **The exact nature of the task proposed in the paper is unclear. What exactly is the input "abstract" and what do the labels associated with it say about the input?**
>
> The nature of the task is to classify Wikipedia abstracts into labels which are organized in an ontology structure. The labels are the categories assigned to the main subject of the abstracts. For example, the abstract for “Affidavit” is given labels of “Evidence Law”, “Legal Documents”, and “Notary”. We include this description in the revision (line 171-174).
>
> **What is the extent to which models developed with Ontologue will be applicable on these other datasets? Eg would it be possible to obtain meaningful text associated with labels in these datasets?**
>
> We thank you for the question. First, we want to point out that the context of the label hierarchy is not limited to textual form. The context includes all general information and data associated with the label hierarchy, which is lacking in the existing HMC benchmarks. Using text associated with the labels is only one of the ways to leverage the context of the label hierarchy. Second, as we mention in Section 3.1, “node labels have been replaced with integers, and data item features may be pre-processed using outdated feature engineering techniques”. These domains, such as protein function predictions and annotation of medical images do have labels associated with text or other features. We expect the HMC benchmarks to reflect this nature of the problem.

---

> ### Author Response · Authors · 2022-08-23
> **Response to Reviewer tU6w (2/2)**
>
> **What exactly are these design elements? What is an example of cherry picking in the case of Ontologue which is prevented by its design? What are the benchmark properties?**
>
> We intentionally only allow users to control the starting node, the max hop, and the data coverage. These three variables can only regulate high level properties of a dataset. Some examples of cherry picking are: cutting of a specific node/branch and altering graph structure to favor specific model performance.. These are prevented by the design of Ontologue.
>
> **I would highly encourage the discussion of specific existing few-shot benchmark datasets in this paragraph.**
>
> We appreciate the suggestion. In section 4.1, we discuss how the existing HMC benchmarks compare to current few-shot benchmarks (lines 218-227). More specifically, “The percentage of novel classes to all classes in the test set for MiniImagenet, CIFAR-FS, and CUB are 20%, 20%, and 25%, respectively”, while 18 out of 20 existing HMC benchmarks have more than 20% novel labels, as much as 72.9% for Eisen dataset from GO collection! Hence, we make the claim “Some of these HMC benchmark datasets are arguably more difficult than standard few-shot benchmark datasets”.
>
> **What are the prior results? Please summarize if this is referencing results from other papers.**
>
> We thank you for this comment. Surj outperforms other competitive HMC methods on 17 out of 20 datasets by a large margin, while Surj and C-HMCNN split the leadership on proposed benchmarks. We include the citation in the revision (line282).
>
> **Why is performance influenced by tree height?**
>
> This is a great question and an ongoing interest of research for the HMC problem. Our intuitive hypothesis is that the deteriorating performance from the tree height is due to low data coverage and growing specialization in context. As the hierarchy goes deeper, there are less and less data records for the fine-grained categories. It is also more difficult for the models to differentiate highly semantically similar labels.
>
> **How does the proposed dataset connect/differ from entity linking datasets such as MedMentions or TypeNet?**
>
> We thank you for bringing up these two prior works. These two studies are focusing on three tasks: Mention-Level Typing, Entity-Level Typing, and Entity Linking (Murty et al. 2018). The research MenMentions and TypeNet are facilitating is leveraging entity linking to categorize input sentences. Our proposed datasets can be seen as a Mention-Level Typing task with hierarchical constraints: ​​Given a sentence vector, the model assigns multi-label while these labels follow the hierarchy structure. Another distinction is that the categories for MedMentions and TypeNet were derived from the entities included in the sentences, while the categories for DBPedia are assigned directly for the subject of the abstracts.
> While MedMentions and TypeNet might not be perfectly suitable for HMC, we apply Ontologue to MedMentions for two purposes: (1) Demonstrating the generalization of Ontologue. It is simple to implement Ontologue on a different data source. (2) Allowing users to analyze the subgraphs of MedMentions. Unfortunately, the data from TypeNet is unavailable. The implementation code is posted on GitHub.

---

### Official Review · Reviewer_wEbZ · 2022-07-29
**A long-needed toolkit, with a high quality discussion of current limitations, hopefully turns out not being limited to DBpedia.**

**Rating:** 9
**Confidence:** 5
**Clarity:** Yes.

**Strengths:**

As a person who also spent a significant amount of time on tackling problems with established multi-label data sets, their label distribution, how some of them are mostly single-label or multi-class problems and should not be evaluated in the multi-label field, I found it very satisfying that the authors provide a view on shortcomings of data sets that are currently used. Shortcomings that may confuse the modeler to take artifacts for progress. It is a strong contribution of the work, although in the appendix. Comparing the figures 3 and 4 in the appendix is the main reason for accepting this paper.

Additional strengths of this paper:
- taking the application context and looking at triviality or impossibility of labels in Figure 2.
- discussion of the HMC vs few-shot learning
- using a selection of widely used methods for baseline establishment



**Weaknesses:**

A strong weakness of the contribution is that at the time of review, and this review is a little late, the code and data are still not available in the GitHub repository. It is therefore impossible to assess whether the authors actually deliver on what they write.

Another weakness is that with DBpedia labels, quality is uncertain, a lot depends on which field one chooses and the choice of the hierarchy depth. The problem is especially with recall, as the taxonomy on Wikipedia evolves, it takes time for articles to be reclassified by editors, not to mention new articles to be added to specific labels. The visual analysis method introduced in the paper, helps to identify deterioration as it starts to happen, but it still does not remove the problem that there will be cases of topic + depth combinations where the DBpedia data is not going to allow generating a good HMC data set. I would've welcomed a limitation section discussing the limits of DBpedia in this regard.

**Additional Feedback:**

The need for a toolkit like Ontologue is clear. Should the code delivered by the authors turn out to be well-documented, easy to use, abstract enough to allow running it on other data sources than Wikipedia/DBpedia, this would be a significant proposition. It is unfortunately impossible to assess at the moment of review as the code is not available.

In my opinion, this paper could be located within the top 15% papers, and that is my verdict in the end, but I make it by trusting the authors that they actually deliver on what they write, without being able to verify that.

**Correctness:**

It is impossible to verify the soundness of the data set, as it was not made available. The described approach is sound to the limitation of DBPedia quality, but should produce a correct data set on a quality-checked data source.

**Documentation:**

With the data and code not made available to the relievers, unfortunately, it is impossible to discuss data organization and availability.



**Ethics:**

1-2. The data set surely contains personally identifiable information or information about individuals, as wikipedia contains such information, but as a community we are accepting the ethical standard of Wikipedia in this regard.
3. Wikipedia also contains bias, however it also puts significant effort into limiting bias and in this regard, the community also tends to trust Wikipedia in that.
4. No human experimentation was performed in this work.
5. Wikipedia has not been discredited by the creators at the time of writing this review.

The method is general enough that it will require ethical consideration when deployed and the users should be made aware of this in my opinion, while the authors did not write an ethical statement for the work and did not mention limitations of this kind in the paper. For example, the hierarchy of medical phenomena in DBpedia may be far from completed and also bugged by non-expert annotation errors, compared to evidence-driven, expert constructed hierarchies like ICD.

**Relation To Prior Work:**

The paper differs from prior work by

**Summary And Contributions:**

The authors propose a tool for generating hierarchical multi-label classification benchmark data sets, discuss the issues with currently available solutions, generate their own example benchmark datasets and provide baselines for them using well-established methods.

---

> ### Author Response · Authors · 2022-08-23
> **Response to Reviewer wEbZ**
>
> We thank the reviewer for the positive feedback and the thorough consideration.
>
> **A strong weakness of the contribution is that at the time of review, and this review is a little late, the code and data are still not available in the GitHub repository.**
>
> We apologize for the delay. The code and documentation have been updated.
>
> **Another weakness is that with DBpedia labels, quality is uncertain, a lot depends on which field one chooses and the choice of the hierarchy depth…. I would've welcomed a limitation section discussing the limits of DBpedia in this regard….the authors did not write an ethical statement for the work and did not mention limitations of this kind in the paper**
>
> We agree with your assessment and we include a discussion regarding the bias and the limitations of DBPedia in Section 6 in the revision.
>
> **hopefully turns out not being limited to DBpedia..**
>
> We appreciate your suggestion on generalizing Ontolgue to other data sources. We have applied Ontologue to MedMentions (Mohan 2019 et al.) to demonstrate the generalization of Ontologue. The code of implementation is posted on Github. We also describe this feature in the revision (line 201-204). We hope this allows users for more general studies.

---

> > ### Comment · Reviewer_wEbZ · 2022-08-26
> > **The response meets my hopes**
> >
> > I am satisfied with your response, and I am happy to uphold my score, thank you.

---

### Official Review · Reviewer_PQRc · 2022-07-31
**NeurIPS 2022 Track Datasets and Benchmarks Paper241**

**Rating:** 6
**Confidence:** 2
**Correctness:** Okay
**Clarity:** Okay

**Strengths:**

1. The proposed Ontologue can provide customizable benchmarks.

2. This paper try to evaluate the proposed method from different aspects.

**Weaknesses:**

1. In the introduction part, the paper mentioned "We analyze 20 HMC benchmark datasets...", but in the results part (both in table 1 and figure 2), there are 6 benchmarks are included.

2. This paper compared the four new benchmarks derived by Ontologue with 6 existing benchmarks in section 5 Comparative Analysis of Benchmarks. But the domains of the four new benchmarks are different form the 6 existing benchmarks. Therefore, it is not clear whether the difference of the performance is caused by the benchmarks themselves or caused by the classification problems (domains).

3. In table 3, only one method is used to evaluate the effectiveness of contextual information. It is better to include more methods to evaluate if the contextual information can help to improve the performance for more methods.



**Additional Feedback:**

Please refer to the weak points above.

**Documentation:**

Good

**Relation To Prior Work:**

I am not quite sure but it seems different from prior work.

**Summary And Contributions:**

1. This paper analysis the weaknesses of current HMC benchmarks: the statistical and structural properties not adequately represent the problem space and can confound performance analysis among competing methods.

2. This paper propose a toolkit – Ontologue, to declaratively construct and analyze a subset from DBpedia as a benchmark for HMC problems with indirect control over structural and statistical features.

3. This paper perform comparative analysis of the new benchmarks against the old benchmarks, and evaluate the performance of state of the art models using four new benchmarks derived by Ontologue.

---

> ### Author Response · Authors · 2022-08-23
> **Response to Reviewer PQRc**
>
> We thank you for the comments. (All mentions of the paper content indicate the first submission unless specified)
>
> **In the introduction part, the paper mentioned "We analyze 20 HMC benchmark datasets...", but in the results part (both in table 1 and figure 2), there are 6 benchmarks are included.**
>
> As we mention in Section 4.1 (line 206-209), all FUNCATdatasets share the same label hierarchy and all GO datasets also share the same label hierarchy. While many papers report on a variety of these "sub" datasets, they are semantically and structurally similar. Due to space limit, we used “cellcycle” subset to represent FUNCAT and GO collections. With the additional space for the revision, we include the whole analysis in Table 1 in the revision.
>
> **It is not clear whether the difference of the performance is caused by the benchmarks themselves or caused by the classification problems (domains).**
>
> We agree that it is unclear what is the cause of the difference in performance. This is still an open problem and Ontologue along with the proposed benchmarks will facilitate studies for this problem. The user of Ontologue will be allowed to control specific aspects of the HMC problem, such as hierarchy depth and data coverage, and study their relationships to the performance.
>
> **In table 3, only one method is used to evaluate the effectiveness of contextual information. It is better to include more methods to evaluate if the contextual information can help to improve the performance for more methods.**
>
> We thank the reviewer for this suggestion. Unfortunately, the current off-the-shelf HMC algorithms do not consider contextual information from the label hierarchy except for Surj. It would be a dramatic change in their models if we re-design for this purpose. One of the main contributions of this paper is to raise the awareness and the importance of the context of the ontologies/hierarchies. We encourage future HMC studies to leverage this concept.
>
> **The organization and the writing of this paper is not good enough. For example, the caption of Figure1 (A) is missing.**
>
> We thank the reviewer for this comment. We do have a caption for Figure 1 (A), “An interactive graph visualization affording qualitative review of the contextual and structural properties of the derived label hierarchy.” We would appreciate more specific suggestions for the writing and the organization.

---

> > ### Comment · Reviewer_PQRc · 2022-08-29
> > **The response meets my hope**
> >
> > I appreciate the authors' efforts in making the paper clearer and the available GitHub repository, I'd like to increase the rating.

---

### Review · Ethics_Reviewer_VehU · 2022-08-21

**Recommendation:** 1

**Ethics Documentation:**

All datasets were obtained from DBPedia. In the paper they say that this is under the Creative Common license although there are several other licenses listed on the DBPedia website. There is no documentation on maintenance.

**Ethics Review:**

Several concerns were raised due to the lack of available code/data for review and the variation of quality in DBpedia which the paper relies on. Since reviewing, the code and data has been published to the GitHub repo.

---

### Author Response · Authors · 2022-08-13
**Code for Ontologue is available!**

We apologize for the delay. The code for Ontologue is available on https://github.com/seanyang38/Ontologue.

We thank Reviewer tU6w to bring up MedMentions and Reviewer wEbZ's feedback to generalize Ontologue. We include code to apply Ontologue to different sources of data.

We will provide detailed comments on each review in the next few days.

---

### Author Response · Authors · 2022-08-23
**Summary of the Revision**

We thank all the reviewers for comprehensive comments and feedback.

We have updated our draft based on the reviews and the followings are the summary of the edits:

1. We added a figure to illustrate the pipeline of Ontologue for clarity.
2. We added a paragraph for the nature of the task (line 171-174)
3. We added a paragraph for the capacity of Ontologue to generalize to other data sources (line 201 - 204)
4. We added a paragraph for the bias and limitation of DBPedia and Wikipedia in section 6.

We appreciate your review and we look forward to your response soon.

---

### Meta-Review · Area_Chair_Jzs6 · 2022-09-08

**Recommendation:** Accept
**Confidence:** 4

**Metareview:**

This paper proposed a benchmark for hierarchical and ontological multi-label classification. Most reviewers find that the paper is interesting and reveals some important shortcomings of the existing multi-label classification datasets. During the rebuttal period, the authors' response addressed most concerns raised by reviewers. I recommend acceptance.

---

### Decision · Program_Chairs · 2022-09-16

Accept